# Properties of SBS/MCF-Modified Asphalts Mixtures Used for Ultra-Thin Overlays

Quanman Zhao [1,*,†] , Xiaojin Lu [1,†], Shuo Jing [1], Yao Liu [1], Wenjun Hu [1,*], Manman Su [2], Peng Wang [1], Jifa Liu [3], Min Sun [1] and Zhigang Li [1]

1 School of Transportation Engineering, Shandong Jianzhu University, Jinan 250101, China; luxiaojin0329@163.com (X.L.); jingshuo99@163.com (S.J.); liuyao199704@163.com (Y.L.); wangpeng@sdjzu.edu.cn (P.W.); sunmin@sdjzu.edu.cn (M.S.); lizhigang@sdjzu.edu.cn (Z.L.)
2 School of Civil Engineering, Yantai University, Yantai 264005, China; ldusuman@126.com
3 Tai'an Highway Planning and Design Institute, Tai'an 271000, China; believevae@163.com
* Correspondence: zhaoquanman@sdjzu.edu.cn or bestcupid@163.com (Q.Z.); huwenjun@sdjzu.edu.cn (W.H.)
† These authors contributed equally to this work.

**Abstract:** In order to produce high-viscosity and high-toughness asphalt for ultra-thin overlays, the conventional asphalt cement was modified with high-content SBS and micro carbon fiber (MCF). The performances of the modified asphalt were studied by tests of penetration, softening point, ductility, kinematic viscosity, multiple stress creep recovery (MSCR), and by dynamic shear rheometer (DSR) and bending beam rheometer (BBR). Mixture properties were studied by tests of rutting, low-temperature bending, freeze–thaw splitting, four-point bending fatigue and dynamic modulus. The results reflect that the addition of MCF could enormously improve the high- and low-temperature properties, increase the viscosity of asphalt, and improve the toughness of asphalt. When SBS content was 6%, with the increase of MCF content, $G^*/\sin \delta$ and R values first increased and then decreased, and the $J_{nr}$ value first decreased and then increased. When MCF content was 0.8%, the overall performance was best. Adding MCF into an asphalt mixture or increasing the content of SBS improved the rutting resistance, low-temperature crack resistance, water stability, and fatigue performance of the asphalt mixture. At the same temperature and frequency, there was little difference in phase angle between the 6%SBS + 0.8%MCF and 5%SBS + 0.0%MCF modified asphalt mixtures, and the dynamic modulus was slightly higher over the whole range. It can be concluded that the addition of SBS and MCF can enormously enhance the viscosity and toughness of asphalt. The viscosity of the 6%SBS + 0.8%MCF modified asphalt met the performance requirements of high-viscosity asphalt. When used for ultra-thin overlays, it had great road service performance and met the application requirements.

**Keywords:** road engineering; ultra-thin overlays; micro carbon fiber; modified asphalt; road performance

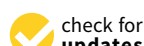



## 1. Introduction

As of 2020, the total highway mileage in China measured at up 5.20 million kilometers, including 0.16 million kilometers of expressway. A total of 98.96% of highway is maintained, and the maintenance task is arduous. In order to restore the pavement function, extend the pavement's life, and save building materials [1], ultra-thin overlays are often used as a new technology for preventive maintenance [2]. It is suggested in the "Technical Specifications for Maintenance of Highway Asphalt Pavement" (JTG 5142-2019) that high-viscosity modified asphalt, rubber asphalt, and high-content SBS-modified asphalt can be used for ultra-thin overlays [3]. Therefore, so as to promote the use of ultra-thin overlays in preventive maintenance of asphalt pavement, new high-performance modified asphalt needs to be researched and developed.

In the 1970s, France first proposed the use of thin overlays and ultra-thin overlays in the maintenance of asphalt pavement, and Nova Chip with a skeletal void structure

was used for ultra-thin overlays in the 1980s [4]. In recent years, so as to heighten the performance of ultra-thin overlays, many scholars have begun to focus on improving the performance of asphalt [5–8]. Chen [9] and Zhou et al. [10] added SBS and a tackifier into a conventional asphalt cement and found that the modified asphalt had higher viscosity and better high-temperature performance than the conventional asphalt cement. A new SMA-5 high-viscosity asphalt mixture was developed by Ren et al. [11]; its performance is better than SBS-modified asphalt mixture, particularly in high-temperature stability and fatigue life. Liu et al. [12] appraised the high-temperature property of polyphosphoric acid–modified asphalt and proposed that the high-temperature property could be accurately evaluated by softening point and cumulative strain. Li et al. [13] used small penetration-grade asphalt to produce high-viscosity asphalt and found that the modification process could greatly improve the rutting resistance and reduce the sensitivity to loading frequency. Zhang et al. [14] developed a new type of SBS-PU high-viscosity asphalt and compared its properties with those of two kinds of high-viscosity modified asphalt, SINOTPS and TPS. Other scholars prepared high-viscosity bitumen by adding rubber powder into SBS [15]; Zhou [16] and Yang et al. [17] analyzed its thermal storage stability and viscoelasticity, and concluded that it has the best performance when the content of rubber powder is 30%. Ming et al. [18] evaluated the rheological properties of several kinds of high-viscosity modified asphalt and proposed that the appropriate modifier should be selected according to the actual needs.

As the most widely used asphalt admixture, fiber materials [19] have the functions of absorbing oil, increasing viscosity, and enhancing toughness. Their type and content directly affect the asphalt properties, for the sake of affecting the high- and low-temperature performance, fatigue durability, and water stability of an asphalt mixture. Micro carbon fiber (MCF) is a kind of crystalline graphite material with more than 95% carbon content and a 0.007–0.044 mm fineness. Thanks to its excellent properties [20–24], it is a widely used material. Its excellent electrochemical properties allow it to be commonly used in the energy field [25]. It is often added to composites to improve the interlaminar properties because of its good toughening effect [26]. It is widely used as a reinforcement material [27] in view of its low density, high modulus [28], and high strength. After mixing it into asphalt, it produces a bridging effect and forms a unique interface construction with the asphalt, effectively limiting the drift of the asphalt and enhancing the high-temperature rheological properties of the asphalt so as to reduce rutting diseases, reduce the temperature reactivity of the asphalt, and improve its low-temperature crack resistance [29]. Dong et al. [30] added biochar into asphalt and found that its aging performance, rutting resistance, and low-temperature performance were improved. Liu et al. [31] studied carbon fiber–graphite-modified asphalt and found that alterations of internal resistance of the pavement can be determined by its conductivity, so as to predict pavement cracking. Vo et al. [32] verified that the tensile strength of an asphalt mixture was significantly improved after adding carbon fiber. Hasan et al. [33] studied the property of an asphalt modified with carbon nanofibers and it was found that all of the performance properties were improved, especially the adhesion of the asphalt.

In short, there are many mediums for ameliorating the performance of asphalt but few studies on adding MCF into asphalt. In this paper, SBS and MCF were added to the conventional asphalt cement for the purpose of achieving modification. The characteristics of the modified asphalt were appraised by tests of penetration, softening point, ductility, kinematic viscosity, multiple stress creep recovery (MSCR) and by dynamic shear rheometer (DSR) and bending beam rheometer (BBR). Mixture properties were studied by tests of rutting, low-temperature bending, freeze–thaw splitting, four-point bending fatigue, and dynamic modulus. The influence of MCF and SBS dosage on the performance of the modified asphalts was analyzed, and the optimal dosages of SBS and MCF were determined. Then, by studying the pavement performance of the SBS/MCF composite modified asphalt mixtures, their applicability in ultra-thin overlays was analyzed.

## 2. Materials and Methodology

### 2.1. Materials

2.1.1. Raw Materials

The raw materials required for the article are MCF, styrene butadiene styrene (SBS), Qinhuangdao 70# conventional asphalt cement, additive (including stabilizer and extracted oil), and aggregates. The major technical indexes of the new type of fiber material MCF are presented in Table 1. SBS is the linear T6302 produced by Dushanzi Petrochemical. WDJ4H stabilizer and Iranian extracted oil were selected. The aggregates are basalt gravel with particle sizes of 5–10 mm and 0–3 mm. Table 2 presents the relevant technical indicators of the aggregate and mineral powder. Table 3 presents the basic indexes of the conventional asphalt cement.

**Table 1.** Technical indexes of MCF.

| Index | Data Results |
|---|---|
| Specification type(mm) | 0.017 |
| Density (g/cm$^3$) | 1.8 |
| Tensile strength (MPa) | 4900 |
| Modulus of elasticity (GPa) | 230 |

**Table 2.** Technical indexes of basalt aggregate and mineral powder.

| Material Type | Relative Apparent Density | Relative Gross Volume Density | Water Content (%) | Asphalt Absorption Coefficient | Relative Effective Density (g/cm$^3$) |
|---|---|---|---|---|---|
| Basalt 5–10 | 2.9307 | 2.8194 | 1.3511 | 0.60 | 2.8859 |
| Basalt 0–3 | 2.8689 | 2.8689 | - | 0.93 | 2.8689 |
| Mineral Powder | 2.6335 | 2.6335 | - | - | 2.6335 |

**Table 3.** Indexes of conventional asphalt cement.

| Technical Property | Unit | Test Results | Specification Requirement |
|---|---|---|---|
| Penetration (25 °C) | 0.1 mm | 63.8 | 60~80 |
| Penetration index (PI) | - | −1.47 | −1.5~+1.0 |
| Ductility (10 °C) | cm | 52 | ≮20 |
| Softening point | °C | 47 | ≮46 |
| Flash point (opening) | °C | 320 | ≮260 |
| Kinematic viscosity (60 °C) | Pa. s | 198 | ≮180 |
| Density (15 °C) | g/cm$^3$ | 1.032 | ≮1.01 |
| After Thin-Film Oven Test (TFOT) | | | |
| Mass Loss | % | 0.11 | ≯±0.8 |
| Penetration (25 °C) | 0.1 mm | 42.3 | - |
| Residual penetration ratio | % | 66.3 | ≥61 |
| Residual ductility | cm | 10.5 | ≮6 |

2.1.2. Mixture Composition and Production

1.  Modified Asphalt

The modified asphalt was prepared according to a specific process [12]. The content of SBS was 5%, 6%, and 7%, the content of MCF was 0.0%, 0.4%, 0.8%, and 1.2%, the content of extracted oil was 2%, and the content of stabilizer was 0.2%.

## 2　Modified Asphalt Mixture

Owing to the requirements of ultra-thin overlays, SMA-10 was selected as the grading type. According to the maximum and minimum values of gradation specified in the specification [34], within the scope of the design specification [35]. Figure 1 displays the gradation of SMA-10.

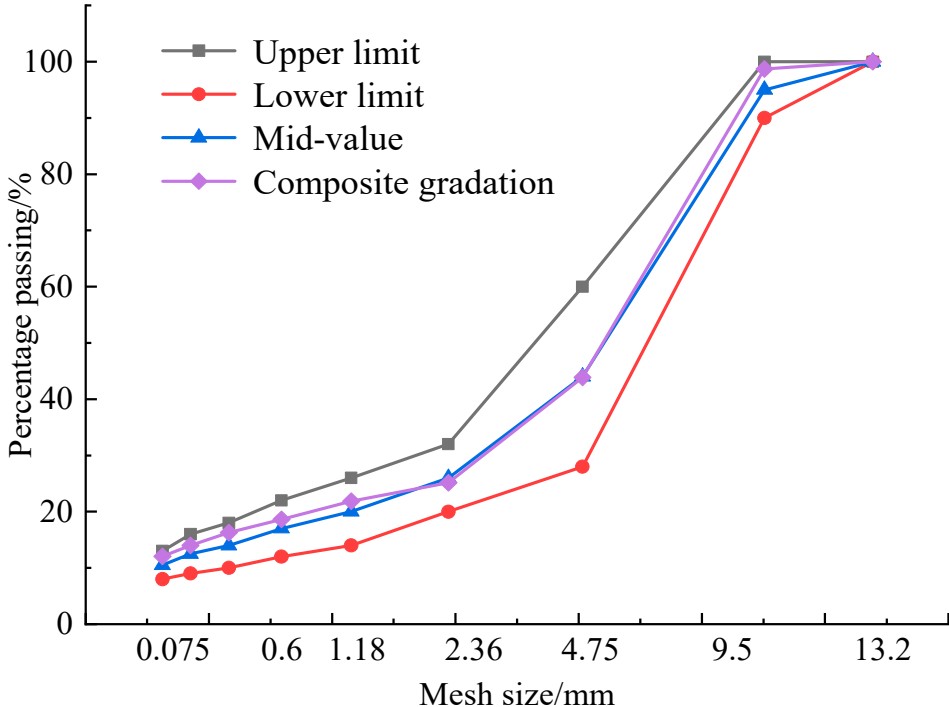

**Figure 1.** Designed gradation curve.

After the gradation of SMA-10 was designed, Marshall specimens were geared up and tested in light of the specification requirements [35], and the most appropriate asphalt content was 6.8%. Table 4 presents the Marshall indexes of SMA-10 obtained through the Marshall test, which all meet the specification requirements [35].

**Table 4.** SMA-10 Marshall specimen test results.

| Marshall Indexes | Gross Volume Density (g/cm³) | Maximum Theoretical Density (g/cm³) | Volume of Air Void (VV/%) | Voids in Mineral Aggregate (VMA/%) | Voids Filled with Asphalt (VFA/%) | Stability (kN) | Flow Value (mm) |
|---|---|---|---|---|---|---|---|
| Test Results | 2.452 | 2.555 | 4.0 | 20.3 | 80.1 | 16.5 | 4.9 |
| Specification Requirements | - | - | 3~4 | ≥17.0 | 75~85 | ≥6.0 | - |

### 2.2. Methodology

2.2.1. Tests for Modified Asphalt

(1)　Conventional tests

Penetration, softening point, ductility, 60 °C kinematic viscosity, and viscosity toughness tests were conducted in accordance with "Standard Test Methods of Bitumen and Bituminous Mixtures for Highway Engineering" (JTG E20-2011) [35].

(2)　Rotary viscosity

The rotary viscosity test was performed in keeping with the specification [35]. The measured temperatures were 100 °C, 120 °C, 135 °C, 155 °C, and 175 °C, and the rotor model selected was 27#.

(3)    DSR

The temperature range of asphalt sample scanning was 52~88 °C, the heating rate of DSR was 2 °C/min, the strain of 27# rotor was 0.5%, and the frequency was 1 Hz. The variation law of rutting factor $G^*/\sin\delta$ at different temperatures was measured.

(4)    MSCR

MSCR was carried out continuously at 64 °C and the stress levels were 0.1 kPa and 3.2 kPa [13]. Each stress level was maintained for 10 cycles, and each cycle [36] lasted 10 s (1 s loading creep stage and 9 s unloading recovery stage).

(5)    BBR

Preparation and test operation of BBR were carried out according to specification [35] and American AASHTO M320-10. The test temperatures were −12 °C, −18 °C, and −24 °C [30].

2.2.2. Tests for Asphalt Mixture

(1)    Rutting test

Rutting test was conducted at 60 °C according to the specification [35].

(2)    Low-temperature bending test

According to the specification [35], beams with 250 mm × 30 mm × 35 mm were prepared as the test specimens, and the single-point loading was implemented at the temperature of −10 °C with the loading speed of 50 mm/min.

(3)    Freeze–thaw splitting test

Freeze–thaw splitting test was conducted according to the specification [35].

(4)    Four-point bending fatigue test

Stress control mode was chosen to conduct the four-point bending fatigue test. Therefore, the load was applied at the speed of 0.01 mm/s and the test would not stop until the specimen broke in the static load test. The stress levels were selected at the stress ratios of 0.3, 0.4, and 0.5, a non-discontinuous asymmetric equal amplitude sine wave was used as the loading waveform for this experiment, and the loading frequency was 10 Hz [37].

(5)    Dynamic modulus test

Dynamic modulus test specimens were prepared according to the specification [35], and the test parameters of the gyratory compactor (Cangzhou Huayun Experimental Instrument Co., Ltd., Cangzhou, China) were set as follows: rotation angle 1.25°, rotation rate 30 r/min, and vertical pressure 600 kPa. The dynamic modulus test of the modified asphalt mixture was carried out at 5 °C, 20 °C, 30 °C, 45 °C and the load frequencies were 0.5 Hz, 1 Hz, 5 Hz, 10 Hz, and 20 Hz, respectively. In order to reduce the test error, the loading was applied in the order of load frequency and test temperature, from low to high. Figure 2 presents the complete experimental design of the study.

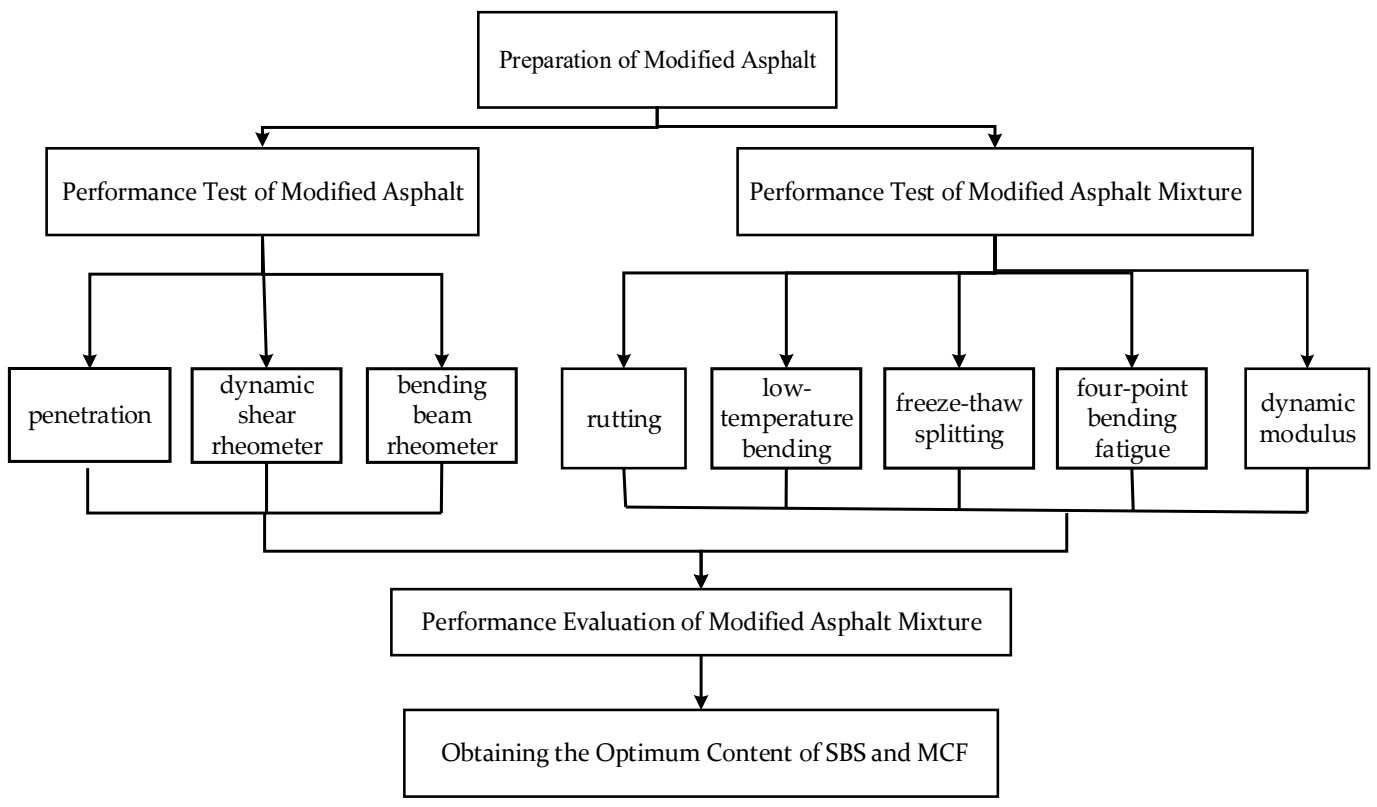

**Figure 2.** The experimental design.

## 3. Results and Discussion

### 3.1. Properties of Modified Asphalt

Table 5 presents the basic index tests results of the modified asphalts.

**Table 5.** Test results of basic indexes of modified asphalts.

| Modified Asphalt Type | Penetration (mm) | Softening Point (°C) | Ductility (mm) | 60 °C Kinematic Viscosity (kPa·s) | Viscosity Toughness (N·m) | Toughness (N·m) |
|---|---|---|---|---|---|---|
| 5%SBS + 0.0%MCF | 53.77 | 74.70 | 29.5 | 16.00 | 24.6 | 16.0 |
| 5%SBS + 0.4%MCF | 52.77 | 76.90 | 30.2 | 19.00 | 27.0 | 21.0 |
| 5%SBS + 0.8%MCF | 52.40 | 77.70 | 33.5 | 22.00 | 30.0 | 22.0 |
| 5%SBS + 1.2%MCF | 53.60 | 74.45 | 34.6 | 20.15 | 30.2 | 23.0 |
| 6%SBS + 0.0%MCF | 52.83 | 77.70 | 31.2 | 17.00 | 27.2 | 18.1 |
| 6%SBS + 0.4%MCF | 51.33 | 78.50 | 32.4 | 23.00 | 29.8 | 22.0 |
| 6%SBS + 0.8%MCF | 50.77 | 79.10 | 34.2 | 25.00 | 32.0 | 23.7 |
| 6%SBS + 1.2%MCF | 51.70 | 79.02 | 35.1 | 22.20 | 34.2 | 24.0 |
| 7%SBS + 0.0%MCF | 48.80 | 81.05 | 32.6 | 21.45 | 32.7 | 20.6 |
| 7%SBS + 0.4%MCF | 46.40 | 81.55 | 33.0 | 29.75 | 34.0 | 24.0 |
| 7%SBS + 0.8%MCF | 46.30 | 82.55 | 34.8 | 31.25 | 34.6 | 24.6 |
| 7%SBS + 1.2%MCF | 47.90 | 81.05 | 35.5 | 28.84 | 35.1 | 25.2 |

According to the results, it can be summarized that increasing the content of MCF while keeping the content of SBS unchanged, with the increase in MCF content, the penetration first decreased and then heightened, the softening point and 60 °C kinematic viscosity first increased and then decreased, and the ductility, viscosity toughness, and toughness gradually increased. When the content of MCF remained unchanged, increasing the content of SBS, the penetration decreased gradually, and the softening point, ductility, 60 °C kinematic viscosity, viscosity toughness, and toughness increased continuously. When the SBS

content was 5%, 6%, and 7%, the best content of MCF was 0.8%, in which the penetration was the minimum, the softening point was the highest and the viscosity toughness was the maximum. The data depict that the addition of MCF could notably advance the high- and low-temperature performance of conventional asphalt cement, significantly enhance the kinematic viscosity, and effectively improve the viscosity toughness. When SBS content was 6% and MCF content was 0.8%, the kinematic viscosity of the modified asphalt was in line with the requirements of ≥20,000 Pa·s for high-viscosity asphalt.

Based on the above analysis, 5%SBS + 0.0%MCF, SBS (content: 5%, 6%, 7%) + 0.8%MCF and 6%SBS + MCF (content: 0.0%, 0.4%, 0.8%, 1.2%) were selected as the optimum asphalt content to research the rheological performances of modified asphalt, and then determine the optimal content of SBS and MCF for ultra-thin overlays.

### 3.2. Rheological Properties of Asphalt

#### 3.2.1. Rotational Viscosity Analysis

Figure 3 presents the test results.

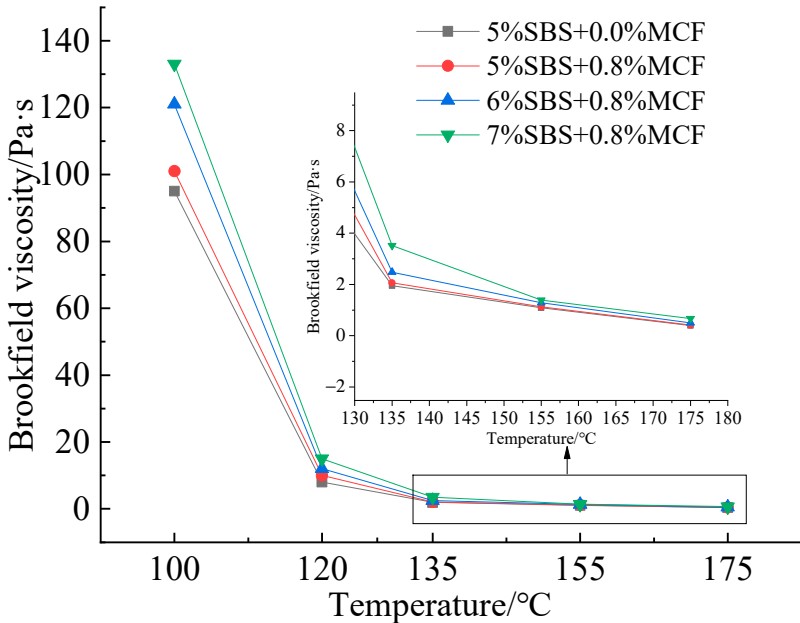

**Figure 3.** Rotational viscosity test result for the modified asphalt.

It can be concluded from Figure 3 that when the content of MCF was constant, the viscosity increased slowly with the increase of SBS content. The viscosity raise significantly after adding MCF, when the content of SBS was 5%, indicating that the viscosity of the modified asphalt was notably heightened by adding SBS and MCF. At all temperatures, the viscosity relationship of asphalt was the following: 7%SBS + 0.8%MCF > 6%SBS + 0.8%MCF > 5%SBS + 0.8%MCF > 5%SBS + 0.0%MCF. At 135 °C, the viscosity of 7%SBS + 0.8%MCF was 3.517 Pa·s (>3 Pa·s), the specification requirements were not met [34]. At 135 °C, the viscosity of 6%SBS + 0.8% MCF was 2.475 Pa·s, and the construction work ability of the modified asphalt met the specification requirements [34].

#### 3.2.2. DSR

DSR test was used to appraise the high-temperature performance of the modified asphalts [14]. Figure 4 reveals the variation law of rutting factor G*/sinδ.

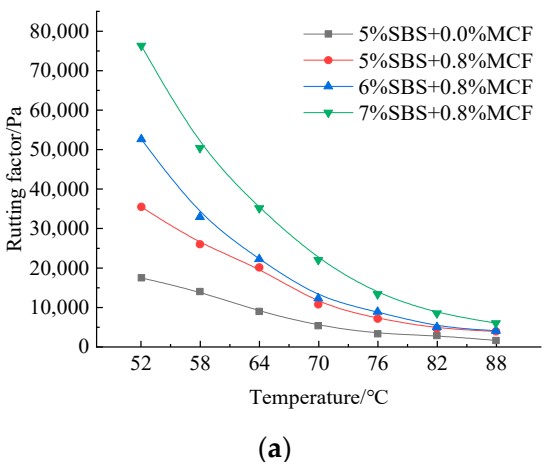
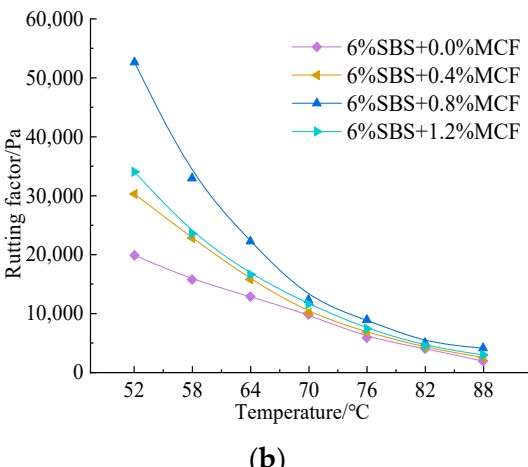

**Figure 4.** Rutting factors for different modified asphalts. (**a**) Rutting factors for the modified asphalts with different SBS content; (**b**) Rutting factors for modified asphalts with different MCF content.

The conclusion can be drawn from Figure 4 that $G^*/\sin\delta$ is coversely associated with the change of temperature and gradually decreases with the expand of temperature, and the higher the temperature, the smaller the difference. Figure 4a shows that $G^*/\sin\delta$ gradually heightened with the increase of SBS content. Figure 4b depicts that when the SBS content was 6%, $G^*/\sin\delta$ increased first and then decreased with the increase of MCF content. When the MCF content was 0.8%, it reached the maximum, and the high-temperature deformation resistance was the best. Taking 64 °C as an example, when the content of MCF was 0.8%, $G^*/\sin\delta$ was increased by 108.5%, 21.6%, and 6.3%, respectively, compared with when the MCF content was 0.0%, 0.4%, and 1.2%.

3.2.3. MSCR

Table 6 presents repeated cumulative strain (S), deformation recovery rate (R), and average unrecoverable creep compliance ($J_{nr}$), as obtained from the MSCR test.

**Table 6.** MSCR test results for the modified asphalts.

| Asphalt Type | 5%SBS + 0.0%MCF | 5%SBS + 0.8%MCF | 6%SBS + 0.8%MCF | 7%SBS + 0.8%MCF | 6%SBS + 0.0%MCF | 6%SBS + 0.4%MCF | 6%SBS + 1.2%MCF |
|---|---|---|---|---|---|---|---|
| $S_{0.1kPa}$ | 0.25 | 0.10 | 0.04 | 0.05 | 0.06 | 0.09 | 0.11 |
| $S_{3.2kPa}$ | 8.67 | 4.32 | 1.30 | 1.90 | 2.12 | 3.00 | 2.48 |
| $J_{nr0.1}$ (kPa$^{-1}$) | 0.25 | 0.10 | 0.04 | 0.05 | 0.06 | 0.09 | 0.08 |
| $J_{nr3.2}$ (kPa$^{-1}$) | 0.27 | 0.13 | 0.04 | 0.06 | 0.07 | 0.09 | 0.11 |
| $R_{0.1}$ (%) | 46.91 | 75.10 | 85.68 | 85.17 | 82.69 | 72.25 | 64.69 |
| $R_{3.2}$ (%) | 47.07 | 70.38 | 84.54 | 82.81 | 79.07 | 70.42 | 82.02 |

The data from Table 6 support that the variation law of cumulative strain with time under the two stress levels was overall the same, and the strain of 6%SBS + 0.8%MCF was not greater than for other modified asphalts, indicating that adding MCF to SBS-modified asphalt improved the deformation resistance, and the performance was the best when the content was 6%SBS + 0.8%MCF. When the SBS content was unchanged, adding MCF could effectively reduce the $J_{nr}$ value of the asphalt binder. When the content of MCF remained unchanged, enhancing the content of SBS, the $J_{nr}$ value under the two stress levels first dropped and then was increased, indicating that increasing the content of SBS could improve the rutting resistance of asphalt, and the performance of 6%SBS + 0.8%MCF was the best.

Under the two stress levels, although the variation law of R was inconsistent at SBS content of 5%, 6% and 7%, R was the largest when the content of MCF was 0.8%. Keeping

the content of MCF unchanged, the value of R under the two stress levels increased gradually with the increase of SBS content. This showed that adding the suggested amount of MCF to the modified asphalt could heighten the deformation resistance of the asphalt binder, and the optimum content was 0.8%.

### 3.2.4. Low-Temperature Bending Test

Low-temperature crack resistance of the modified asphalts was evaluated by gauging the stiffness modulus (S) and the creep rate (m) of the modified asphalts by BBR. Figure 5 shows the test results.

The conclusion can be concluded from Figure 5 that when the temperature was $-12\,°C$ and $-18\,°C$, the values of S and m met the requirements of the specification (S $\leq$ 300 Mpa and m $\geq$ 0.3). All of the modified asphalts had the same low-temperature grade which was PG-28. When the temperature was $-24\,°C$, the modified asphalts had little difference in m value, but there was a lot of difference in S value. Therefore, it was one-sided to use S or m value to dissect the low-temperature property of asphalt [38].Therefore, the Burgers model was used to evaluate the low-temperature performance of asphalt with m/S values, as shown in Figure 6.

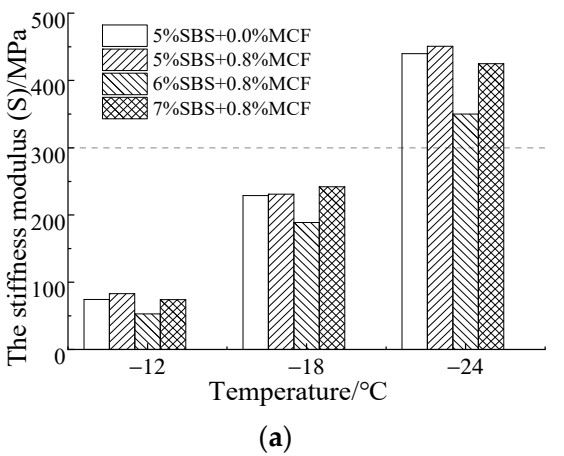
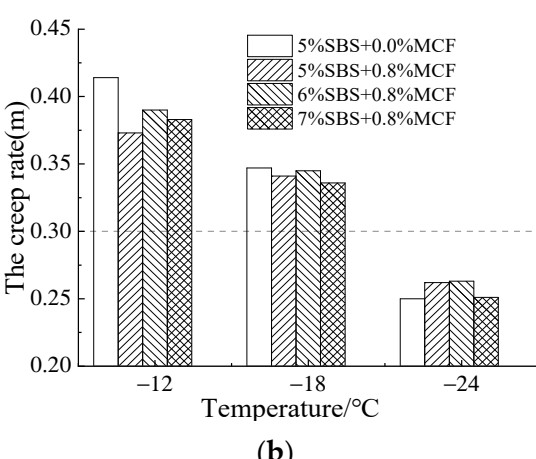

**Figure 5.** BBR results of modified asphalts at different temperatures. (**a**) Test results for S value; (**b**) Test results for m value.

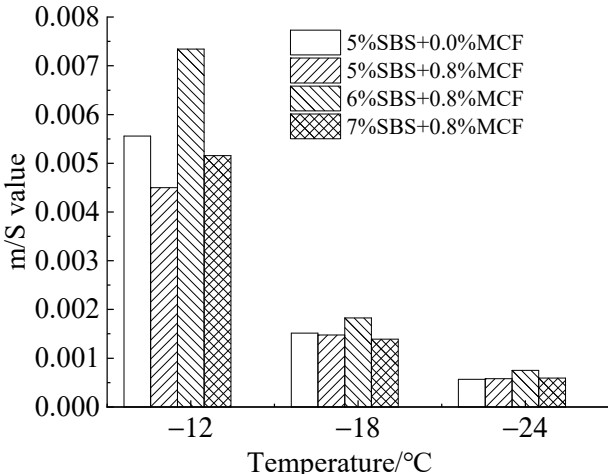

**Figure 6.** Values of m/S for different modified asphalt.

It is clear in Figure 6 that the m/S value of modified asphalt are positively correlated with temperature, and the decline range from $-12\,°C$ to $-18\,°C$ was greater than that from

−18 °C to −24 °C. The differences of m/S values for modified asphalts became smaller when the temperature decreased. The m/S value of 6%SBS + 0.8%MCF was the largest, indicating that its low-temperature performance was the best.

### 3.3. Performance of the Modified Asphalt Mixtures

### 3.3.1. High-Temperature Stability

Table 7 presents the dynamic stability (DS) index of rutting test [35].

**Table 7.** Results of the rutting test.

| Mixture Type | $d_{45min/mm}$ | $d_{60min/mm}$ | DS (time/mm$^{-1}$) | Specification Requirement (time/mm$^{-1}$) |
|---|---|---|---|---|
| 5%SBS + 0.0%MCF | 1.706 | 1.830 | 5045 | |
| 5%SBS + 0.8%MCF | 1.728 | 1.840 | 5625 | ≥2500 |
| 6%SBS + 0.8%MCF | 1.903 | 1.996 | 6774 | |
| 7%SBS + 0.8%MCF | 1.323 | 1.398 | 8362 | |

Table 7 shows that DS values of 5%SBS + 0.8%MCF, 6%SBS + 0.8%MCF, and 7%SBS + 0.8%MCF increased by 11.5%, 20.4%, and 48.7%, respectively, compared with 5%SBS + 0.0%MCF, indicating that enhancing the content of SBS and MCF can effectively increase the high-temperature stability of an asphalt mixture.

### 3.3.2. Low-Temperature Crack Resistance

Table 8 presents the test outcomes of the low-temperature bending tests and specification [34] requirements.

**Table 8.** Results of low-temperature bending tests.

| Mixture Type | Flexural Tensile Strength $R_B$ (MPa) | Failure Strain $\varepsilon_B$ (µε) | Bending Stiffness Modulus SB (MPa) | Specification Requirement (MPa) |
|---|---|---|---|---|
| 5%SBS + 0.0%MCF | 7.532 | 2867 | 2627.0 | |
| 5%SBS + 0.8%MCF | 9.343 | 3226 | 2896.2 | ≥2500 |
| 6%SBS + 0.8%MCF | 10.506 | 3578 | 2936.2 | |
| 7%SBS + 0.8%MCF | 11.375 | 3447 | 3299.9 | |

These results of Table 8 revealed that the failure strain of 6%SBS + 0.8%MCF was the largest, indicating that after the crack appeared, the crack development speed of the specimen was the slowest, and its fracture toughness was better than that of other specimens. For ultra-thin overlays, modified asphalt with strong strain resistance is preferred, so 6%SBS + 0.8%MCF modified asphalt would be preferred for ultra-thin overlays.

### 3.3.3. Water Stability

Table 9 presents the freeze–thaw splitting test results and from it, the order of freeze–thaw splitting strength ratio was established as: 7%SBS + 0.8%MCF > 6%SBS + 0.8%MCF > 5%SBS + 0.8%MCF > 5%SBS + 0.0%MCF, indicating that adding MCF or increasing the content of SBS can improve the water stability of an asphalt mixture.

**Table 9.** Results of the freeze–thaw splitting tests.

| Mixture Type | $\overline{R}_{T1}$ (MPa) | $\overline{R}_{T2}$ (MPa) | TSR (%) | Specification Requirement (%) |
|---|---|---|---|---|
| 5%SBS + 0.0%MCF | 1.492 | 1.352 | 90.6 | |
| 5%SBS + 0.8%MCF | 1.805 | 1.650 | 91.3 | TSR ≥ 80 |
| 6%SBS + 0.8%MCF | 1.695 | 1.585 | 93.5 | |
| 7%SBS + 0.8%MCF | 1.697 | 1.617 | 95.3 | |

### 3.3.4. Fatigue Performance

Through the previous research, it was found that the performance of the 6%SBS + 0.8%MCF modified asphalt mixture was excellent. Therefore, fatigue and dynamic modulus tests were tested on it and it was in conflict with the conventional 5%SBS + 0.0%MCF modified asphalt mixture. A four-point bending test [24,25] was used to test the fatigue property of the asphalt mixture. Figure 7 presents the bending stress–deflection curve for bending tensile strength test. It can be concluded, as summarized in Figure 7, that under static loading conditions, there was little difference between 6%SBS + 0.8%MCF and 5%SBS + 0.0%MCF for the properties of stress deflection. So as to comprehensively judge the fatigue performance of the asphalt mixtures under distinct stress standards, fatigue tests at the stress ratio levels of 0.3, 0.4, and 0.5 were implemented. Figure 8 presents the results of the fatigue test and it can be concluded that the fatigue life decreased significantly with the enhancement of the stress ratio. With a stress ratio of 0.3, 0.4, and 0.5, respectively, the fatigue life of the 6%SBS + 0.8%MCF modified asphalt mixture was 1.17, 1.13, and 1.32 times that of the 5%SBS + 0.0%MCF modified asphalt mixture, respectively, and the fatigue performance of the 6%SBS + 0.8%MCF modified asphalt mixture was excellent.

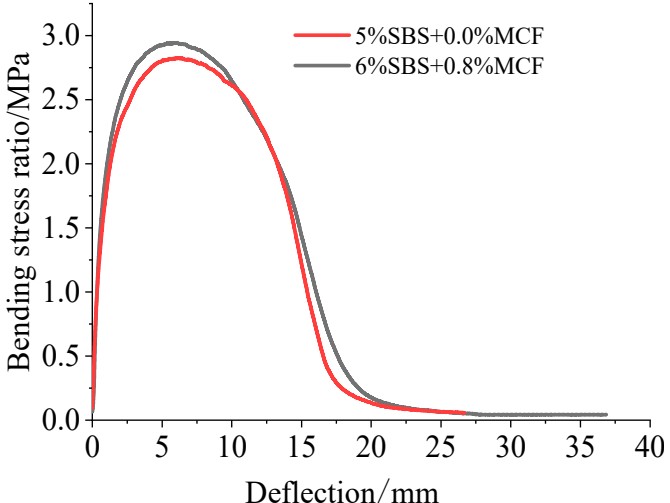

**Figure 7.** Bending stress–deflection curve.

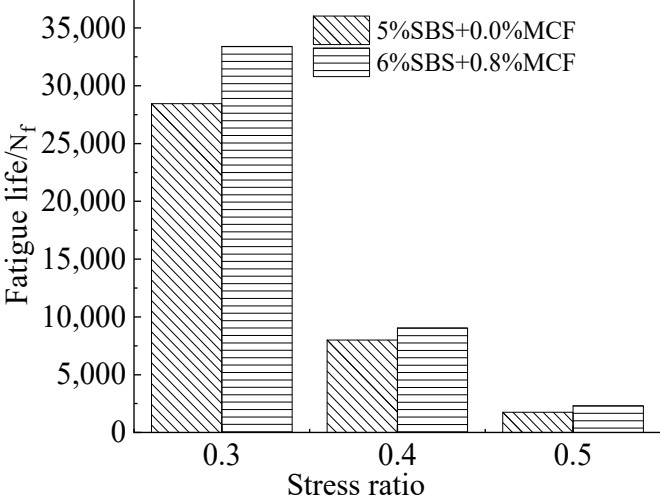

**Figure 8.** Results of fatigue tests for different stress levels.

### 3.3.5. Dynamic Modulus

Tables 10 and 11 present separately the data of the dynamic modulus and phase angle obtained from the dynamic modulus test.

**Table 10.** Results of the dynamic modulus test for modified asphalt mixtures.

| Frequency/Hz | Dynamic Modulus (MPa) | | | | | | | |
|---|---|---|---|---|---|---|---|---|
| | 5%SBS + 0.0%MCF | | | | 6%SBS + 0.8%MCF | | | |
| | **5 °C** | **20 °C** | **30 °C** | **45 °C** | **5 °C** | **20 °C** | **30 °C** | **45 °C** |
| 0.1 | 5526.5 | 1341.0 | 596.6 | 307.0 | 6133.0 | 1486.0 | 646.4 | 339.5 |
| 0.5 | 7906.0 | 2284.0 | 919.5 | 393.3 | 8781.0 | 2529.5 | 992.7 | 427.9 |
| 1 | 9085.5 | 2918.5 | 1185.0 | 458.9 | 9923.0 | 3227.5 | 1279.5 | 492.1 |
| 5 | 12007.0 | 4775.0 | 2118.5 | 793.1 | 13014.0 | 5251.0 | 2295.0 | 832.5 |
| 10 | 13264.5 | 5795.5 | 2713.0 | 995.8 | 14387.5 | 6352.0 | 2947.0 | 1047.5 |
| 20 | 14359.0 | 6915.0 | 3488.0 | 1293.5 | 15814.5 | 7555.0 | 3776.5 | 1377.5 |

**Table 11.** Results of phase angle for modified asphalt mixtures.

| Frequency/Hz | Phase Angle (°) | | | | | | | |
|---|---|---|---|---|---|---|---|---|
| | 5%SBS + 0.0%MCF | | | | 6%SBS + 0.8%MCF | | | |
| | **5 °C** | **20 °C** | **30 °C** | **45 °C** | **5 °C** | **20 °C** | **30 °C** | **45 °C** |
| 0.1 | 24.260 | 30.130 | 25.395 | 18.600 | 23.085 | 29.950 | 25.155 | 17.830 |
| 0.5 | 20.435 | 30.640 | 29.120 | 21.775 | 19.555 | 30.150 | 29.270 | 22.160 |
| 1 | 18.805 | 29.700 | 30.015 | 23.910 | 17.760 | 29.090 | 30.275 | 24.630 |
| 5 | 14.835 | 26.985 | 31.195 | 26.565 | 13.930 | 25.960 | 31.245 | 27.880 |
| 10 | 13.310 | 25.370 | 30.910 | 28.610 | 12.520 | 24.270 | 30.800 | 29.965 |
| 20 | 11.955 | 23.515 | 29.740 | 30.005 | 11.245 | 22.420 | 29.550 | 31.315 |

(1) The relationship between dynamic modulus and temperature

As is shown in Table 10, during high-frequency loading, the reduction range of dynamic modulus gradually decreased as the temperature rose. On the contrary, it increased with the decrease of temperature during low-frequency loading. Under the same conditions frequency or temperature, comparing the dynamic modulus of the 6%SBS + 0.8%MCF and 5%SBS + 0.0%MCF, it was found that the former had a larger value. This result indicates that 6%SBS + 0.8%MCF could maintain a higher modulus at higher temperatures, and had stronger resistance to high-temperature rutting than that of 5%SBS + 0.0%MCF modified asphalt pavement.

(2) The connection between phase angle and temperature

Table 11 presents that, under the same temperature and frequency, comparing the phase angle of the 6%SBS + 0.8%MCF and the 5%SBS + 0.0%MCF, the value of the former is slightly lower. This indicates that 6%SBS + 0.8%MCF had better elasticity, and better resistance to high-temperature rutting.

(3) Establishment of master curve

For viscoelastic materials, the time–temperature conversion principle was available for use in studying the mechanical properties under different temperatures and load frequencies. The master curve was established through Formula (1) [39], and the obtained master curve parameter values are shown in Table 12. The main curve parameters were brought into Equations (2) and (3) to obtain the master curve equation for dynamic modulus and shift factor at different temperatures, so as to draw the master curve of dynamic modulus and master curve of phase angle [40], as presented in Figure 9.

$$\log|E*| = \delta + \frac{Max - \delta}{1 + e^{\beta + \gamma \log t_r}} \tag{1}$$

$$\log[a(T)] = \frac{\Delta E_a}{19.14714} \left( \frac{1}{T} - \frac{1}{T_r} \right) \tag{2}$$

$$\log|E*| = \delta + \frac{Max - \delta}{1 + e^{\beta + \gamma \{\log f + \frac{\Delta E_q}{19.14714} [(\frac{1}{T}) - (\frac{1}{T_r})]\}}} \tag{3}$$

where $|E^*|$—Dynamic modulus (MPa); *Max*—Logarithm of ultimate maximum dynamic modulus; $t_r$—Reduction time at reference temperature; $\beta$, $\gamma$, $\delta$—Fitting parameters, and the initial value is $\delta = 0.5$, $\beta = -1.0$, $\gamma = -0.5$.

**Table 12.** Parameter values for master curve.

| Mixture Type | $\delta$ | $\Delta E_a$ | $\beta$ | $\gamma$ | $S_e/S_y$ | $R^2$ |
|---|---|---|---|---|---|---|
| 5%SBS + 0.0%MCF | 4.3693 | 200000 | 0.4020 | 0.5976 | 0.0044 | 0.999 |
| 6%SBS + 0.8%MCF | 4.4550 | 200000 | 0.4134 | 0.6377 | 0.0039 | 0.999 |

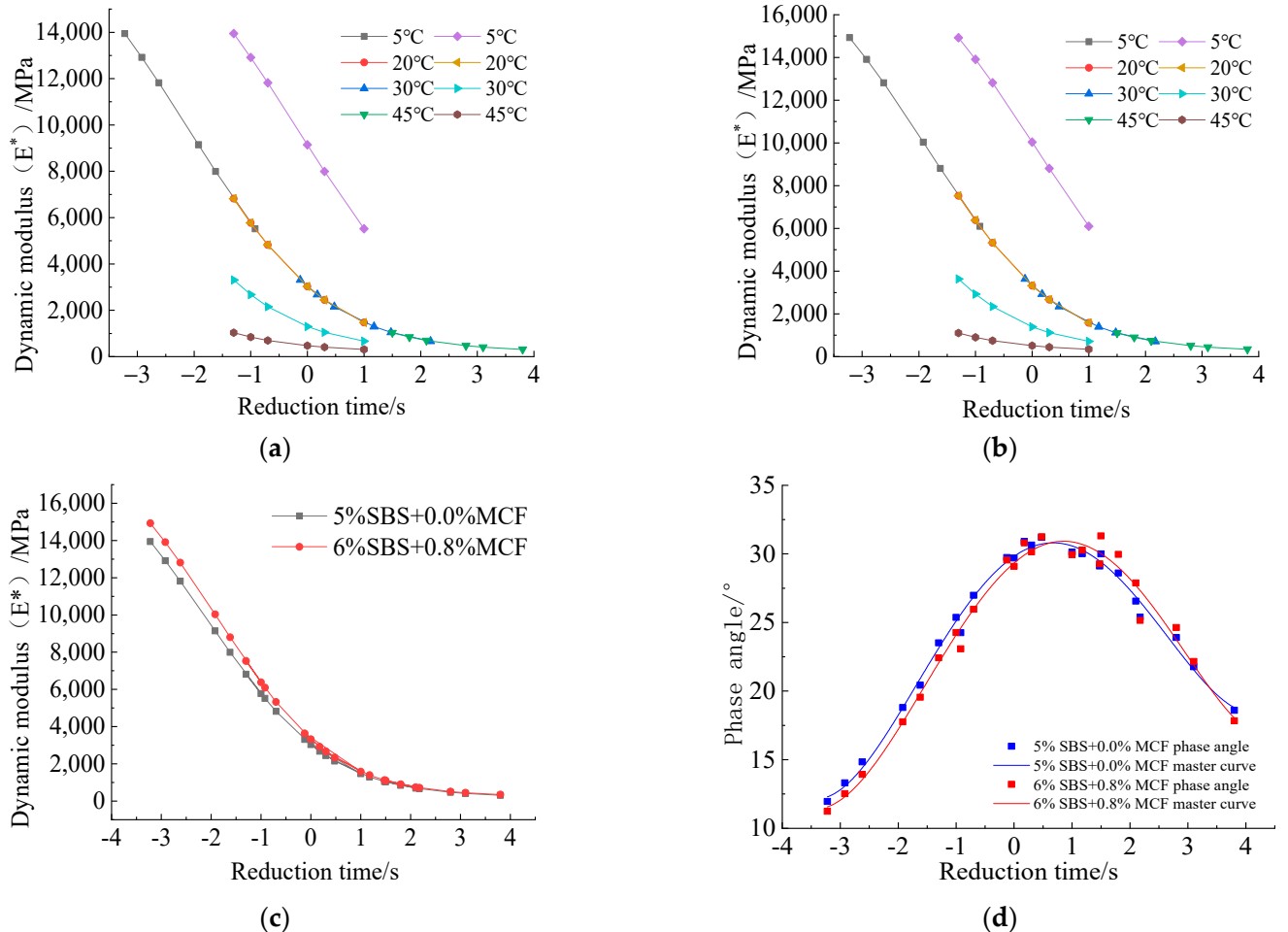

**Figure 9.** Master curve for modified asphalt mixtures. (**a**) Master curve for 5%SBS + 0.0%MCF; (**b**) Master curve for 6%SBS + 0.8%MCF; (**c**) Comparison of master curves for dynamic modulus; (**d**) Comparison of master curves for phase angle.

As indicated in Figure 9a,b, the dynamic modulus decreased with the increase of reduction time, with a slower trend at both ends of the curve and an accelerated decline in the middle of the curve. Figure 9c shows that the influence of frequency on the 6%SBS + 0.8%MCF and 5%SBS + 0.0%MCF asphalt mixtures was overall the same, compared with 5%SBS + 0.0%MCF asphalt mixture, 6%SBS + 0.8%MCF mixture has a higher dynamic modulus, indicating that the deformation resistance of the 6%SBS + 0.8%MCF

modified asphalt mixture was better compared to the 5%SBS + 0.0%MCF asphalt mixture. In Figure 9d it can be seen that the phase angle of the two asphalt mixtures had a tiny gap, indicating that the content of SBS and MCF had no significant effect on the phase angle.

## 4. Conclusions

The major conclusions are as follows:

(1) Addition of MCF can obviously heighten the high- and low-temperature properties of asphalt, significantly increase the kinematic viscosity, and effectively improve the viscosity and toughness of asphalt. The addition of SBS and MCF can lead to high viscosity and toughness in asphalt. The kinematic viscosity of the 6%SBS + 0.8%MCF modified asphalt met the criteria for high-viscosity asphalt ($\geq$20,000 Pa·s);

(2) DSR and MSCR experiments showed that when the SBS content was 6%, with the increase in MCF content, the $G^*/\sin\delta$ and R values first increased and then decreased, and the $J_{nr}$ value first decreased and then increased. The optimum content of MCF was 0.8%. Adding an appropriate amount of MCF into high-content SBS-modified asphalt can significantly heighten the high-temperature deformation resistance. BBR tests found that a high content of SBS and an appropriate content of MCF can enhance the low-temperature performance of asphalt, and the best high- and low-temperature performance of modified asphalt was achieved when the SBS content was 6% and the MCF content was 0.8%;

(3) It was found that adding MCF or increasing the content of SBS in modified asphalt can increase the rutting resistance, low-temperature crack resistance, water stability, and fatigue performance of the asphalt mixture, and the performance of the 6%SBS + 0.8%MCF modified asphalt mixture was the best, which is feasible for ultra-thin overlays.

(4) The dynamic modulus test indicated that for the phase angle of the 6%SBS + 0.8%MCF and 5%SBS + 0.0%MCF modified asphalt mixture there was little difference at equal temperature and frequency. The dynamic modulus of the 6%SBS + 0.8%MCF asphalt mixture was superior to that of the 5%SBS + 0.0%MCF asphalt mixture. The modified asphalt mixture with 6%SBS + 0.8%MCF had better deformation resistance than the 5%SBS + 0.0%MCF mixture.

The modified asphalt mixture of 6%SBS + 0.8%MCF not only had good viscosity and toughness, but also its road performance was better than that of general modified asphalt. Therefore, it can be used in road maintenance technology as ultra-thin overlays.

**Author Contributions:** Conceptualization, Q.Z. and W.H.; methodology, Q.Z., M.S. (Manman Su) and M.S. (Min Sun); software, X.L.; validation, Q.Z. and Y.L.; formal analysis, X.L. and W.H.; investigation, Y.L., S.J. and Z.L.; resources, Q.Z. and P.W.; data curation, X.L.; writing—original draft preparation, X.L.; writing—review and editing, Q.Z. and W.H.; visualization, X.L.; supervision, Q.Z. and W.H.; project administration, Q.Z.; funding acquisition, Q.Z. and J.L. All authors have read and agreed to the published version of the manuscript.

**Funding:** This research was financially supported by the project 2021B24 supported by Shandong Transportation Science and Technology Plan, and project ZR2020QE273 supported by Shandong Provincial Natural Science Foundation.

**Institutional Review Board Statement:** Not applicable.

**Informed Consent Statement:** Not applicable.

**Data Availability Statement:** Using the data of this article requires application from the author.

**Acknowledgments:** We express our sincere gratitude to the experts, teachers, and students who have provided help for this paper.

**Conflicts of Interest:** The authors proclaim no conflict of profit.

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
