# Peer review of "Properties of SBS/MCF-Modified Asphalts Mixtures Used for Ultra-Thin Overlays"

_coatings, doi:10.3390/coatings12040432_

Round 1

Reviewer 1 Report

Points 1: Page 1, Abstract -What do you meant with "matrix”asphalt? Conventional asphalt cement? If yes, substitute by conventional asphalt and apply to the whole article. If not define "matrix"asphalt when it appears for the first time.

Thanks for your suggestion. Matrix asphalt refers to the conventional asphalt cement used to produce modified asphalt. Matrix asphalt was substituted by conventional asphalt cement and applied to the whole article.

Points 2: Page 3 , Section 2.1.1.- Use SI units (mm) to translate the "mesh grid dimension". Do the same in the whole article.

Thanks for your improvement suggestion, for your proposal to use the SI units(mm) to translate the "mesh grid dimension" has been implemented.

Points 3: The research plan expressed in this article was fairly defined and accomplished. However, the conclusions were not well established. In fact, the asphalt 6%SBS+0.8%MCF does not significantly improve the mechanical properties of the SMA produced compared with the one that uses 5%SBS+0%MCF. And this for all the checked properties. Yes, there are differences and better answers for the mixture using 6%SBS+0.8%MCF, but the amplitude of those differences is tiny, and for sure, the modified asphalt has a higher cost than the one with just 5% of SBS. So, my advice is to discuss better the differences and indicate why you believe that the indicated asphalt modification is keener to be used in practice. Adding some production cost comparisons will also help the reader to have a better reference to understand the whole picture.

Thanks for your recognition of our work. For your questions, I will answer from the following aspects. Ultra-thin overlay is a preventive maintenance measure for asphalt pavement. In Chinese specifications Technical Specifications for Preventive Maintenance of Highway Asphalt Pavement, it is recommended to use high viscosity modified asphalt (the kinematic viscosity of the modified asphalt met the requirements of ≥20000 Pa•s for high viscosity asphalt.) and high content SBS modified asphalt for hot mix ultra-thin overlay. And through reading literature, it is found that carbon fiber modified asphalt often shows superior mechanical properties. Based on the above two factors, it is considered to add micro carbon fiber(MCF) into SBS modified asphalt to improve the performance of asphalt, and judge whether the SBS/MCF modified asphalts is suitable for ultra-thin overlay.

Reviewer 2 Report

Points 1: The abstract and conclusion should highlight the novelty of the current study.

Thanks, I’ve improved it. I have pointed out at the beginning and end of the abstract and the third conclusion that the 6%SBS+0.8%MCF modified asphalt mixture studied in this paper is suitable for repairing the damage of pavement in ultra-thin overlay.

Points 2:Research gaps should be appropriately addressed in the introduction.

Thanks for your kindly advice on this article, I’ve improved it. In the last paragraph of the introduction, I explained that based on carbon fiber, micro carbon fiber (MCF) and SBS modified asphalt are selected for composite modification, and whether the performance of the asphalt mixture meets the requirements for ultra-thin overlay.

Points 3: The complete experimental design for the study is missing, along with any flow chart if possible for better understanding.

Thanks for your suggestion. We have made a test flow chart and put it on page six of the article. Please check it and make suggestions.

Points 4: The binder hardens after short-term aging from 60-70 to 40-50 which is significant, all the tests are performed on the original condition not considering the effect of binder after aging. If the effect on properties is as shown for the short-term aging than the effect of long term aging will be much greater so the pavement constructed using these materials will be more prone to fatigue cracking. Considering the fact it is an overlay, cracking in the overlay will not be good for the underlying layer.

Thanks for your questioning the test process of this article. The test process in this article is carried out in accordance with the Standard Test Methods of Bitumen and Bituminous Mixtures for Highway Engineering. For the selected base asphalt, after the film oven aging test, the asphalt performance meet the specification requirements. The test results show that the selected 6%SBS+0.8%MCF asphalt mixture has good low-temperature crack resistance and fatigue performance, and it is not easy to produce cracks as the overlayer.

Points 5: The figures need to improve as they selected percentages of SBS and MCF are confusing.

Thanks for your improvement suggestions. For the confusing percentages of SBS and MCF in the article, we have been written uniformly in the form of #%SBS+#%MCF.

Points 6: The comparison of mixture type is not proper, there are gaps in the experimental design.

Thanks for your question. In Chinese specifications Technical Specifications for Preventive Maintenance of Highway Asphalt Pavement, it is recommended to use SMA-5/10 or AC-5/10 hot mix asphalt mixtures or warm mix asphalt mixtures matching the paving thickness for ultra-thin overlays. Combined with engineering experience and a large number of literatures, it is found that asphalt mastic macadam mixture (SMA) is superior to other structural types in high and low temperature performance and water damage resistance, and this structural type of mixture is widely used in high-grade roads, so the type of mixture selected in this paper is SMA-10.

Points 7: Figure 7 the fatigue life/time is not clear. Clarify the units.

Thanks for your improvement suggestions.Through literature review for the fatigue life unit in Figure 7, it has been remarked.

Round 2

Reviewer 2 Report

The authors have sufficiently improved the manuscript based on my comments